# A Fast Registration Method for Optical and SAR Images Based on SRAWG Feature Description

**Zhengbin Wang, Anxi Yu \*, Ben Zhang, Zhen Dong and Xing Chen** 

College of Electronic Science, National University of Defense Technology (NUDT), Changsha 410073, China
\* Correspondence: yu_anxi@nudt.edu.cn; Tel.: +86-0137-8731-7450

**Abstract:** Due to differences in synthetic aperture radar (SAR) and optical imaging modes, there is a considerable degree of nonlinear intensity difference (NID) and geometric difference between the two images. The SAR image is also accompanied by strong multiplicative speckle noise. These phenomena lead to what is known as a challenging task to register optical and SAR images. With the development of remote sensing technology, both optical and SAR images equipped with sensor positioning parameters can be roughly registered according to geographic coordinates in advance. However, due to the inaccuracy of sensor parameters, the relative positioning accuracy is still as high as tens or even hundreds of pixels. This paper proposes a fast co-registration method including 3D dense feature description based on a single-scale Sobel and the ratio of exponentially weighted averages (ROEWA) combined with the angle-weighted gradient (SRAWG), overlapping template merging, and non-maxima suppressed template search. In order to more accurately describe the structural features of the image, the single-scale Sobel and ROEWA operators are used to calculate the gradients of optical and SAR images, respectively. On this basis, the $3 \times 3$ neighborhood angle-weighted gradients of each pixel are fused to form a pixel-wise 3D dense feature description. Aiming at the repeated feature description in the overlapping template and the multi-peak problem on the search surface, this paper adopts the template search strategy of overlapping template merging and non-maximum suppression. The registration results obtained on seven pairs of test images show that the proposed method has significant advantages over state-of-the-art methods in terms of comprehensive registration accuracy and efficiency.

**Keywords:** optical and SAR images; co-registration; consistent gradient calculation; angle-weighted gradient; dense feature description





## 1. Introduction

Synthetic aperture radar (SAR) is an active microwave sensor. Compared with optical sensors, it can realize all-weather earth observation without being affected by weather changes [1]. However, compared to optical images, the imaging properties of SAR images make them not easy to interpret. The high-efficiency and high-precision automatic co-registration of optical and SAR images is a prerequisite for the fusion of their advantages [2]. The co-registration of optical and SAR images is mainly used in image fusion [1], image segmentation [3], change detection [4,5], 3D reconstruction [6], and so on. Image registration is the process of aligning two or more images corresponding to the same scene acquired at different times, different sensors, or under different conditions [7]. The main difficulties encountered in realizing high-performance co-registration of optical and SAR images are: (1) The imaging modes of the two are quite different. (2) There exists a nonlinear intensity difference (NID) between the two images. (3) Multiplicative speckle noise on SAR images makes methods suitable for optical image registration perform poorly when applied to SAR images. A number of researchers have proposed solutions to the above-mentioned problems. At present, optical and SAR image co-registration methods are mainly divided into manual-design-based methods and deep-learning-based methods [8]. Among them,

manual-design-based methods are divided into two types: intensity-based methods and feature-based methods.

Intensity-based methods typically employ template matching. First, a suitable similarity measure is selected for the two images. Then, by selecting a template on the reference image, traversing the search area on the sensed image, and finding the maximum position of the similarity measure, image registration can be completed. According to the different similarity measures, such methods can be divided into normalized cross-correlation (NCC) [9,10], mutual information (MI) [11–13], frequency-domain-based [14], and other methods. The advantage of intensity-based methods is that the algorithm design is simple. The disadvantage is that the search surface for similarity measures is often not smooth and prone to mismatches since these methods are processed directly based on image intensities. Intensity-based methods have difficulty in achieving rotational invariance and are accompanied by the problem of high computational overhead (especially MI).

Compared with intensity-based methods, feature-based methods can reduce the effects of NID and noise to a certain extent, and so they are more suitable for optical and SAR image co-registration. The main processing steps of feature-based methods include feature extraction, feature description, and feature matching. Feature-based methods are mainly divided into two categories: global-feature-based approach and local-invariant-feature-based approach [2]. The features extracted by the global-feature-based methods are line features [15], contour features [16], and shape features [17]. Although these features have certain invariance, not all images have such features, and so their scope of application is limited. The main extracted features based on local invariant features are divided into point features [18–20] and regional features [21]. Among them, since the point features are more common in the image and it is easier to determine their position, the registration methods based on the point features are the more popular method at present.

In the field of computer vision, the scale-invariant feature transform (SIFT) [22] is the most widely applicable image registration method. However, due to the aforementioned problems in optical and SAR image co-registration, SIFT cannot be directly applied to optical and SAR image co-registration. At present, many researchers have proposed improved SIFT methods for the problems existing in optical and SAR image co-registration. Fan et al. believed that the first octave of extracted feature points in the Gaussian pyramid was unstable, which directly skipped the feature extraction of the first octave and did not perform the main direction assignment to improve the uniqueness of feature descriptors [23]. Dellinger et al. introduced the ROEWA operator into the Harris–Laplace detector for corner detection and proposed a gradient ratio (GR) operator to calculate the gradient of the SAR image, obtaining the SAR-SIFT [24]. Gong et al. used SIFT with outlier removal for coarse registration and MI for fine registration and achieved good registration results [25]. Based on the unified level set segmentation model, Xu et al. improved the SIFT and reduced the probability of mismatch through an iterative method [26]. Ma et al. proposed a new gradient based on image intensity difference and enhanced the feature matching method in the feature matching stage to improve the performance of the SIFT [27]. Xiang et al. used multi-scale Sobel and ROEWA operators to calculate the gradients of optical and SAR images, respectively. The locations of the feature points are optimized by the least squares method to improve the performance of the feature point detector [7]. Yu et al. combined the ROEWA operator and nonlinear diffusion filtering to construct an image scale space and used two different improved detectors to detect feature points on optical and SAR images, respectively. Finally, a rotation-invariant descriptor based on the Log-Gabor filter was designed [2]. The aforementioned improved methods based on SIFT often require the gradient information of the image to construct descriptors. Some studies in recent years have shown that the phase consistency information can resist the NID between images. Ma et al. performed feature point extraction from the phase consistency map, adding spatial constraints in the feature matching stage to improve registration performance [28]. Fan et al. proposed a Harris feature point detector based on uniform nonlinear diffusion (UND-Harris), which uses phase consistency to construct feature de-

scriptions to achieve co-registration of optical and SAR images [29]. Liu et al. combined the Gabor filter and maximum stable extremal region (MSER) to achieve affine invariance of descriptors and proposed the maximum stable phase consistency (MSPC) descriptor [30]. Li et al. performed feature point extraction by phase-consistent maximum and minimum moments and performed feature description with a maximum index map (MIM), acquiring a radiation-invariant feature transform (RIFT) descriptor with rotation invariance [31].

Although the aforementioned feature-based methods can provide good scale and rotation invariance, for mature commercial remote sensing data products, sensor imaging parameters are usually provided by the supplier in the image file, which can realize the unified conversion of the image space of optical and SAR images to the geographic coordinate space. Therefore, the coarse registration of the two images can be completed by rational polynomial coefficients (RPCs) or the range–Doppler (RD) model. The coarse registration can eliminate the main rotation and scale differences between two images. However, after coarse registration, the relative positioning accuracy between the two images can still reach tens or even hundreds of pixels. In response to this, related researchers have proposed hybrid registration methods combining template matching strategy and feature description. Ye et al. proposed two 3D feature descriptors: the histogram of orientation phase consistency (HOPC) and the channel feature of orientation gradients (CFOG). Then, template matching was used to complete the fine registration of the coarsely registered optical and SAR images [32,33]. Li et al. proposed the HOPES structure feature descriptor using a multi-scale and multi-directional Gabor filter combined with the main edge fusion strategy and also used template matching to complete the feature matching process [34].

Benefiting from the rapid development of deep neural networks, the research on remote sensing image registration based on deep learning has also made great progress in recent years. Ye et al. fused SIFT and deep learning features for the registration of remote sensing images [35]. Ma et al. proposed a two-step registration method for multimodal images, which combines local features and deep learning features [36]. Li et al. proposed a deep neural network to estimate the rotation angle between images and used Gaussian pyramid local feature descriptors to complete remote sensing image registration [37]. Merkle et al. used generative adversarial networks (GANs) to convert optical images into SAR-like images and further transformed the optical and SAR image co-registration problem into a SAR-like and SAR image co-registration problem [38]. Zhang et al. built a general multimodal image registration framework based on Siamese fully convolutional networks [39]. Soon after, they still used Siamese fully convolutional networks to learn deep features of image pixels, improving the robustness of the network [40]. Quan et al. improved the GAN and designed a generative matching network (GMN) to improve registration performance [41]. Bürgmann et al. pre-trained a convolutional neural network (CNN) with medium-resolution images and then fine-tuned high-resolution images to learn common descriptor representations [42]. Cui et al. proposed a convolutional network with a spatial pyramid pooling and attention mechanism, which can overcome complex geometric distortion and NID [43]. Aiming at the problem of an insufficient training set of image data, Hughes et al. constructed an optical and SAR image matching network through semi-supervised learning to overcome the limitation of the dataset [44]. However, although deep learning methods are more likely to exhibit higher performance, they are easily limited by datasets due to the complexity of remote sensing image acquisition conditions (resolution, imaging angle, polarization factors, etc.). There is still a lack of generalization ability. Therefore, manual-based registration methods are still widely used at this stage.

Based on the aforementioned hybrid registration methods, this paper proposes a 3D dense feature description based on Sobel and ROEWA combined with the angle-weighted gradient (SRAWG). Further, we propose a fast template matching method to achieve the fine co-registration of coarsely registered optical and SAR images. Compared with the state-of-the-art methods of the same kind, the main improvements include the following:

1.  A 3D dense feature description based on SRAWG is proposed, combined with non-maximum suppression template search, to further improve registration accuracy. The single-scale Sobel and ROEWA operators are used to calculate the consistent gradients of optical and SAR images, respectively, which have better speckle noise suppression ability than the differential gradient operator. The gradient of each pixel is projected to its adjacent quantized gradient direction by means of angle weighting, and the weighted gradient amplitudes in the $3 \times 3$ neighborhood of each pixel are fused to form the feature description of each pixel. SRAWG can more accurately describe the structural features of optical and SAR images. We introduce a non-maximum suppression method to address the multi-peak phenomenon of search surfaces in frequency-domain-based template search methods. This further improves registration accuracy.

2.  In order to maintain high registration efficiency, we modify the original multi-scale Sobel and ROEWA operators into the single-scale case for the computation of image gradients. For the repeated feature construction problem in template matching, we adopt the overlapping template merging method and analyze the efficiency improvement brought by this method.

The rest of this paper is organized as follows: In Section 2, we present the process of the proposed fast registration method and elaborate on how each processing step is implemented. In Section 3, we verify the effectiveness of the proposed method through experiments and then compare the state-of-the-art methods through comprehensive registration experiments. The effectiveness of our proposed registration method is verified by combining subjective and objective evaluation criteria. In Section 4, we analyze the impact of key parameter settings on registration accuracy and summarize the method's limitations. Finally, the conclusions are presented in Section 5.

## 2. Method

### 2.1. Proposed Fast Registration Method

The flow chart of the fast registration method proposed in this paper is shown in Figure 1 below. The whole process is divided into three main steps: feature point extraction, feature description, and feature matching.

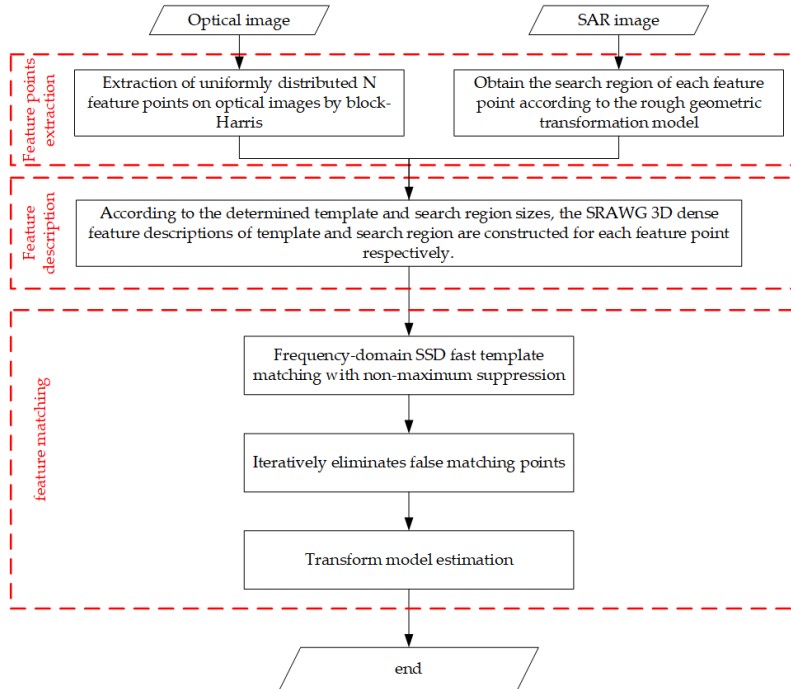

**Figure 1.** The flow chart of the proposed fast registration method.

### 2.2. Implementation Details of Each Step

#### 2.2.1. Feature Point Extraction

This method requires that optical and SAR images need to be coarsely geometrically corrected by the RPC or RD before input. Large scale, rotation, and offset differences between optical and SAR images can be eliminated by a coarse geometric correction step.

For the template matching methods, we need to select an appropriate template region on the reference image for searching. Usually, there are regions with different texture information on an image; for image registration, we only need to focus on regions with significant texture features. Feature points can represent the richness of image texture features, which are mainly divided into corner points and blob points. Since SAR images have rich corner point information, corner points have better performance in optical and SAR image co-registration applications [2]. The traditional Harris corner detector has two main defects:

1. The detector relies on the differential gradient of the image; so when it is directly applied to the SAR image detection task, many false corner points are often obtained.
2. Due to the lack of spatial location constraints, the detected corner points are unevenly distributed in the image space, which has an impact on the final image registration accuracy.

For problem 1, we consider the co-registration requirements of optical and SAR images because template matching is used, and so only feature points need to be detected in one of the two images. Therefore, corner detection on the optical image can avoid false detection problems caused by speckle noise on the SAR image. For problem 2, we can introduce a block strategy to limit the location so that the detected corner points can be evenly distributed in the image space. Figure 2 shows the corner points obtained using the block–Harris detector and Harris detector in an optical image (The number of extracted corner points is 200).

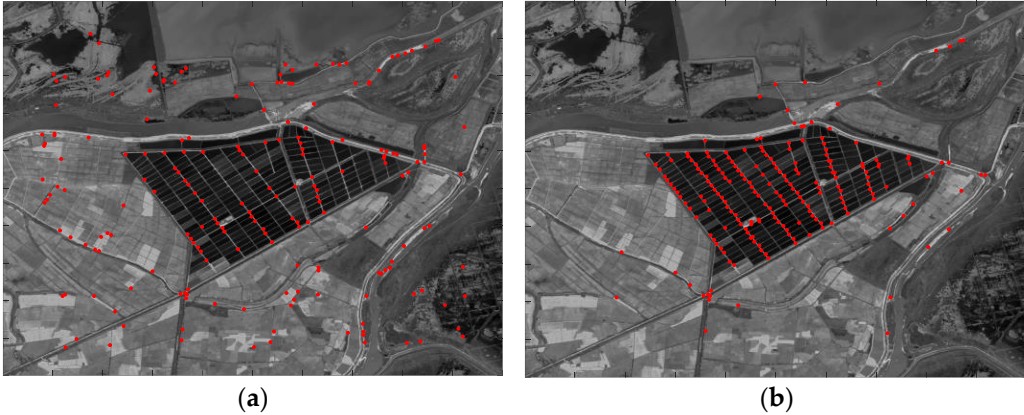

(**a**)　　　　　　　　　　　　　　　(**b**)

**Figure 2.** Comparison of corner extraction results. (**a**) Block–Harris extraction result; (**b**) Harris extraction result.

We can see that when the block strategy is not adopted, the corner points are essentially concentrated in the region with a more significant geometric structure in the middle of the image. After the block strategy is adopted, the corner points can be uniformly distributed on the whole image, which is beneficial to improve the accuracy of image registration. Therefore, in this paper, we choose a block–Harris detector to divide the optical image into $n \times n$ sub-blocks uniformly and detect $k$ feature points in each sub-block. Then, according to the rough geometric transformation model, the position of these feature points on the SAR image is determined as the center of the search region corresponding to each feature point.

### 2.2.2. 3D Dense Feature Descriptions Based on SRAWG

To construct SRAWG 3D dense feature descriptions for optical and SAR images, we need to go through the steps shown in Figure 3.

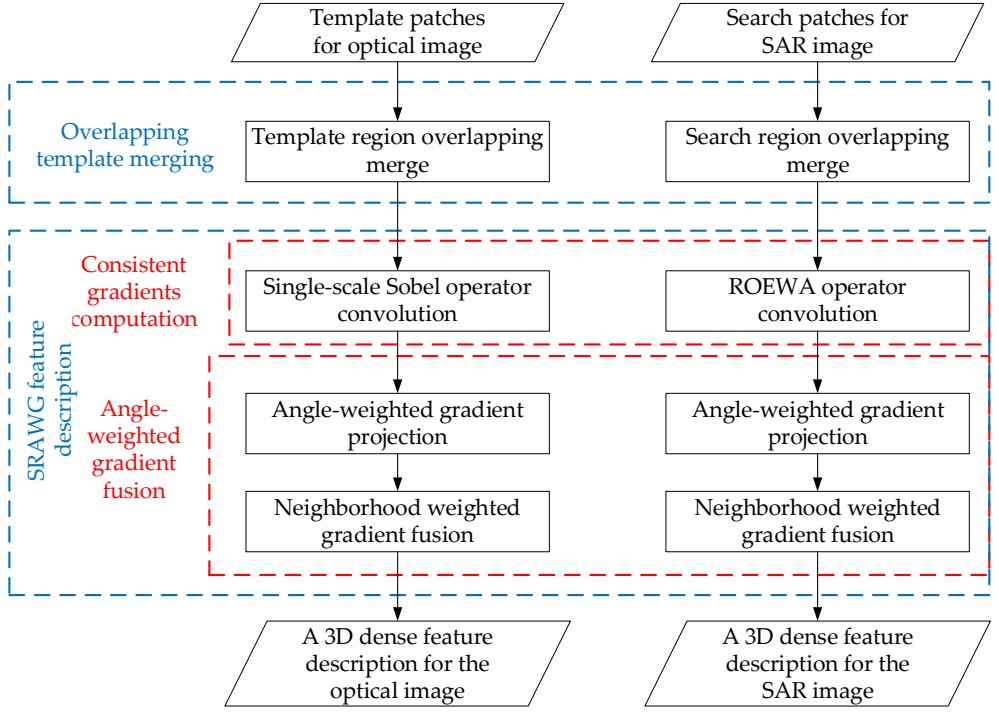

**Figure 3.** SRAWG feature description construction flow chart.

(1) Overlapping template merging

In the traditional template matching process: For the reference image, feature descriptions need to be established in the template region corresponding to the detected feature points. For the sensed image, it is necessary to determine the approximate position of every feature point on the sensed image according to the rough transformation model. Then, establish the feature description of the search region corresponding to each feature point. When we have determined the number of feature points and the size of the template window and search region, the computational complexity of the registration has also been determined. This feature construction method will face the problem of repeated feature descriptions in the overlapping regions of the templates. (For the convenience of description, we collectively refer to the template region of the reference image and the search region of the sensed image as the template region.) In addition, this phenomenon is most serious when the image size is small. In order to reduce the repeated construction of features, a template merging strategy is considered [45].

The specific operation of the template merging strategy is as follows: First, it is necessary to judge whether the sum of the template areas corresponding to the two feature points is greater than the minimum circumscribed rectangle area of the two templates. When the condition of greater than is satisfied, the SRAWG feature description is constructed with the minimum circumscribed rectangular region, and then each feature point in the rectangular region can directly take the corresponding feature description. Figure 4 shows a schematic diagram of the template merging method.

The position coordinates corresponding to the two feature points are $(x_1, y_1)$ and $(x_2, y_2)$, respectively. The red box in the figure corresponds to the template region of the two feature points, the blue box corresponds to the overlapping region of the two templates, and the yellow dotted line box corresponds to the minimum circumscribed rectangle $R$ of the two templates. The coordinates of the four vertices A, B, C, and D corresponding to

the rectangle are $(x_1 - w/2, y_2 - w/2)$, $(x_1 - w/2, y_1 + w/2)$, $(x_2 + w/2, y_1 + w/2)$, and $(x_2 + w/2, y_2 - w/2)$, where $w$ is the template window size.

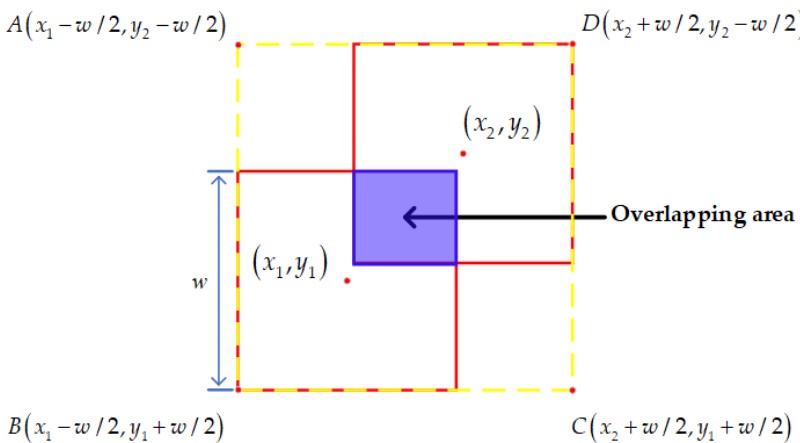

**Figure 4.** Template merging diagram.

If pixel-wise feature descriptions are constructed for the template regions corresponding to the two feature points, the number of feature description vectors to be constructed is $2w^2$. However, if the feature description is constructed with the coverage region of the circumscribed rectangle $R$, the number of feature description vectors that need to be constructed is $s = (x_2 - x_1 + w) \times (y_2 - y_1 + w)$. When $s < 2w^2$ is satisfied, the calculation amount of constructing the feature description with the circumscribed rectangle $R$ is smaller than that of constructing the feature description for each template region separately. We count the sets of all feature points in the image that satisfy the aforementioned relationship. For each set of points, we construct the minimum circumscribed rectangles corresponding to these sets of points, respectively. Finally, we can construct feature descriptions for these circumscribed rectangles.

(2) Consistent gradient calculation of optical and SAR images

How to effectively describe the template and the search region corresponding to each feature point to improve the smoothness of the final search surface is the current research hotspot of template matching methods. Among them, the method of obtaining consistent gradients has made considerable progress: in CFOG, the author uses the differential gradient operator to calculate the horizontal and vertical gradients and then combines sinusoidal interpolation to obtain gradient description with multiple directions. In HOPES, the author uses a multi-scale and multi-directional Gabor filter combined with a main direction fusion strategy to construct a feature description. The former has great advantages in processing efficiency but does not fully consider the sensitivity of the differential gradient operator to speckle noise in SAR images. Although the description method adopted by the latter has obtained a more robust feature description, it is ultimately unsatisfactory in terms of computational efficiency due to the multi-scale feature fusion strategy. In this paper, the single-scale Sobel and ROEWA operators are used to calculate the gradient.

Xiang et al. proposed the use of multi-scale Sobel and ROEWA operators to calculate gradients for optical and SAR images, respectively [7]. The multi-scale Sobel operator is expressed as follows:

$$S_{h,\beta_m} = \mathcal{G}_{\beta_m} \times H_{\beta_m}, \; S_{v,\beta_m} = \mathcal{G}_{\beta_m} \times V_{\beta_m} \tag{1}$$

In the above formula, $S_{h,\beta_m}$ and $S_{v,\beta_m}$ represent the horizontal and vertical gradients, respectively. $\mathcal{G}_{\beta_m}$ denotes a Gaussian kernel with standard deviation $\beta_m$. $H_{\beta_m}$ and $V_{\beta_m}$ denote horizontal and vertical Sobel windows of length $\beta_m$, respectively. $\times$ represents

matrix dot product. Using formula (1) to convolve the optical image, we can obtain the amplitude and direction of the optical image as follows:

$$G_{Om,\beta_m} = \sqrt{\left(S_{h,\beta_m}\right)^2 + \left(S_{v,\beta_m}\right)^2} \tag{2}$$

$$G_{Oo,\beta_m} = \arctan\left(\frac{S_{v,\beta_m}}{S_{h,\beta_m}}\right) \tag{3}$$

For the SAR image, we use the ROEWA operator [46] to calculate the gradient. The ROEWA operator calculates the horizontal and vertical gradient formulas as follows:

$$GR_{x,\alpha_n} = \log(R_{x,\alpha_n}) \tag{4}$$

$$GR_{y,\alpha_n} = \log(R_{y,\alpha_n}) \tag{5}$$

In the above formulas, $GR_{x,\alpha n}$ and $GR_{y,\alpha n}$ represent the horizontal and vertical gradients of the image, respectively, where:

$$R_{x,\alpha_n} = \frac{\sum\limits_{i=-M/2}^{M/2}\sum\limits_{j=1}^{N/2} I(x+i,y+j)e^{-\frac{|i|+|j|}{\alpha_n}}}{\sum\limits_{i=-M/2}^{M/2}\sum\limits_{j=-N/2}^{-1} I(x+i,y+j)e^{-\frac{|i|+|j|}{\alpha_n}}} \tag{6}$$

$$R_{y,\alpha_n} = \frac{\sum\limits_{i=1}^{M/2}\sum\limits_{j=-N/2}^{N/2} I(x+i,y+j)e^{-\frac{|i|+|j|}{\alpha_n}}}{\sum\limits_{i=-M/2}^{-1}\sum\limits_{j=-N/2}^{N/2} I(x+i,y+j)e^{-\frac{|i|+|j|}{\alpha_n}}} \tag{7}$$

where $M$ and $N$ are the length and width of the processing window, respectively. They are closely related to the scale parameter $\alpha_n$, usually taken as $M = N = 2\alpha_n$; $(x,y)$ represents the pixel position of the gradient to be calculated; $I$ represents the pixel intensity of the image.

Further, the magnitude and direction of the gradient can be obtained as follows:

$$G_{Sm,\alpha_n} = \sqrt{\left(G_{x,\alpha_n}\right)^2 + \left(G_{y,\alpha_n}\right)^2} \tag{8}$$

$$G_{So,\alpha_n} = \arctan\left(\frac{G_{y,\alpha_n}}{G_{x,\alpha_n}}\right) \tag{9}$$

To further reduce the effect of gradient reversal between the optical and SAR images, the gradient directions $G_{So,\alpha n}$ and $G_{Oo,\beta m}$ are restricted to [0,180°].

If the optical and SAR images are to be consistent with gradients, the filter effective support area of multi-scale Sobel and ROEWA needs to be the same [47]. Xiang deduced in the article that the two scale parameters $\alpha_n$ and $\beta_m$ of the ROEWA and the multi-scale Sobel operators need to satisfy $\alpha_n = \beta_m$. In this paper, considering that the registration efficiency and the coarse registration eliminate large-scale differences, we modify the multi-scale Sobel and ROEWA operators to the single-scale case. We set the scale parameter to $\alpha_n = \beta_m = 2$.

(3) 3D dense feature description construction based on SRAWG

Compared with the feature description construction by sine interpolation of horizontal and vertical gradients in CFOG, in order to make full use of the gradient information of the pixel neighborhood, we introduce the angle-weighted gradient method mentioned by Fan in [48] to construct a 3D dense feature description. However, unlike in [48], we perform angle weighting on the basis of the extracted optical and SAR image consistent gradients from single-scale Sobel and ROEWA. In addition, we found that after constructing the

preliminary 3D dense feature description, performing 2D Gaussian filtering can improve registration accuracy.

For a certain pixel $(x,y)$ of the image, firstly, it is necessary to quantify the gradient direction of the pixel to obtain $N$ feature directions and project the gradient corresponding to each pixel to its two adjacent quantized feature directions. In this way, we can obtain the corresponding left-weighted gradient magnitude and right-weighted gradient magnitude for each image. Taking an optical image as an example, the schematic diagram of the calculation of two weighted gradients is shown in Figure 5.

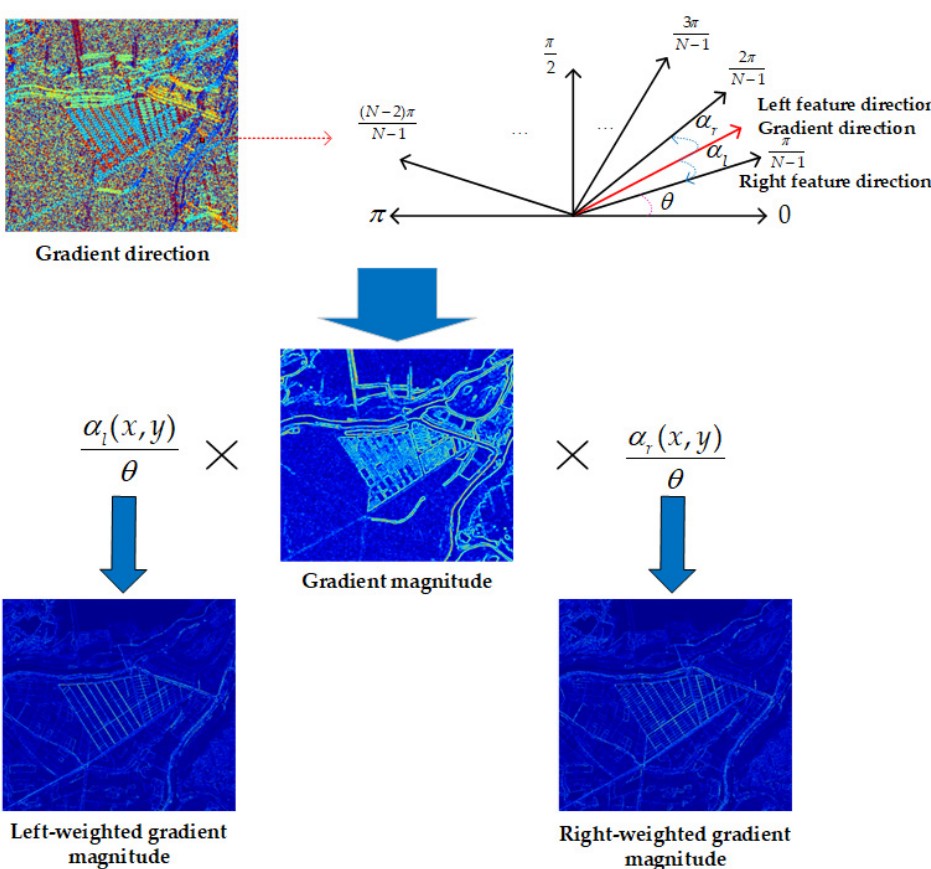

**Figure 5.** Schematic diagram of angle-weighted gradient projection.

We assume that the gradient direction of image pixel $(x,y)$ is $G_o(x,y)$ and the corresponding magnitude is $G_m(x,y)$; then, the gradient projections of this pixel to its right feature direction and left feature direction are:

$$G_{mr}(x,y) = \frac{\alpha_r(x,y)}{\theta} G_m(x,y) \tag{10}$$

$$G_{ml}(x,y) = \frac{\alpha_l(x,y)}{\theta} G_m(x,y) \tag{11}$$

In the above formula, $G_{mr}(x,y)$ and $G_{ml}(x,y)$ are the right-weighted gradient magnitude and the left-weighted gradient magnitude of $G_m(x,y)$, respectively. $\alpha_r(x,y)$ and $\alpha_l(x,y)$ are the angles between the gradient direction of the pixel and its right and left feature directions, respectively. A $\theta$ is the gradient quantization angle interval. Their calculation formulas are as follows:

$$\theta = \frac{\pi}{N-1} \tag{12}$$

$$\alpha_r(x,y) = G_o(x,y) - \left\lfloor \frac{G_o(x,y)}{\theta} \right\rfloor \times \theta \tag{13}$$

$$\alpha_l(x,y) = \theta - \alpha_r(x,y) \tag{14}$$

In the above formulas, $\lfloor \cdot \rfloor$ represents a downward integer fetch operation. That is, when the internal value is positive, it is equal to itself; otherwise, it is 0. After the left-weighted gradient magnitude $G_{ml}(x,y)$ and the right-weighted gradient magnitude $G_{mr}(x,y)$ of the image are obtained, a 3D dense feature description is constructed by characterizing each pixel according to the process shown in Figure 6.

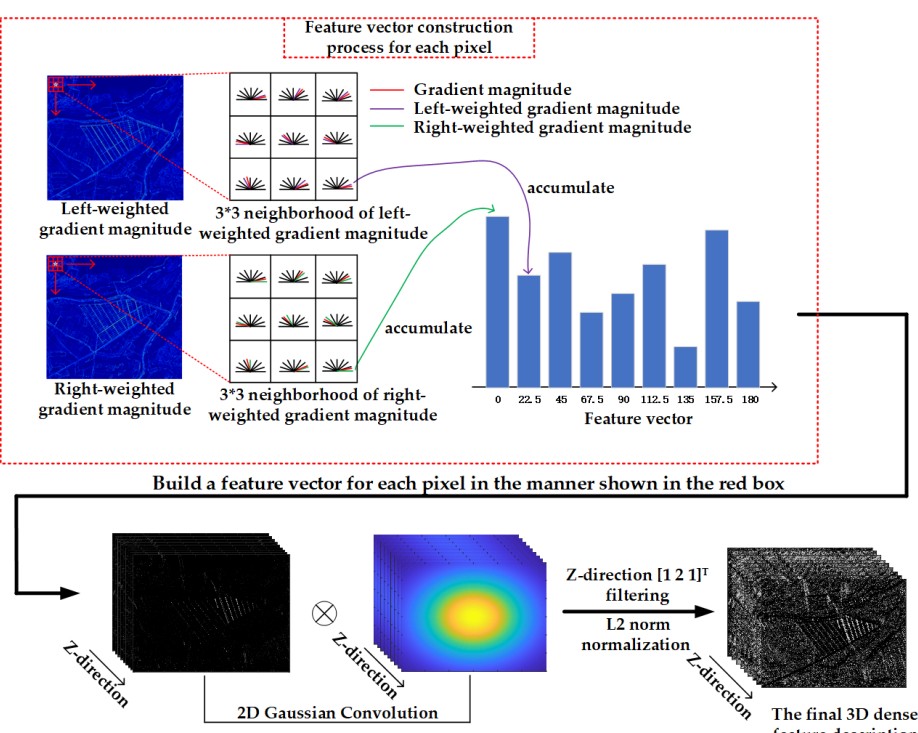

**Figure 6.** Construction schematic diagram of the 3D dense feature description ($N = 9$).

In order to construct the feature description vector at the pixel $(x,y)$, we open a statistical window in the $3 \times 3$ neighborhood around the pixel corresponding to the left-weighted gradient magnitude map and the right-weighted gradient magnitude map. For the quantized gradient directions corresponding to the left feature direction and the right feature direction of these 9 pixels, add the weighted gradients under the same quantized gradient direction so that the initial $N$ dimensional feature vector at pixel $(x,y)$ is obtained.

Construct feature vectors for all pixels in the image in this way, and place the feature vector of each pixel along the Z axis to obtain a preliminary 3D dense feature description. Two-dimensional Gaussian filtering is performed on the corresponding two-dimensional description under each quantized feature direction, respectively. After that, the Gaussian filtered description is filtered along the Z direction with $[1\ 2\ 1]^T$, and the L2 norm normalized is used to obtain the final 3D dense feature description. So far, we have completed the construction of the SRAWG feature description.

### 2.2.3. Feature Matching

In template matching, after the feature descriptions of the template region and the search region are completed, the next step is to perform a sliding search on the search region. The similarity measure between the optical image template and the SAR image template is calculated to obtain the corresponding search surface. By finding the extreme point corresponding to the surface, the matching point coordinates on the SAR image can be obtained. In this paper, we choose the sum of squared differences (SSD) as the similarity measure. Since the constructed 3D dense feature description is pixel-wise, it is time-consuming to compute SSD directly in the spatial domain. Therefore, we refer to the

method in CFOG and use a fast Fourier transform (FFT) to convert the spatial correlation operation of SSD to the frequency domain for point multiplication to speed up feature matching. The SSD calculation formula for feature point $P_i$ is as follows:

$$SSD_i(\Delta x, \Delta y) = \sum_{(x,y)} \left( D^i_{Opt}(x,y) - D^i_{SAR}(x - \Delta x, y - \Delta y) \right)^2 \tag{15}$$

In the above formula, $D^i_{Opt}(x,y)$ and $D^i_{SAR}(x,y)$ represent the corresponding template region 3D feature description on the optical and SAR images, respectively. $SSD_i(\Delta x, \Delta y)$ represents the SSD similarity measure that the template window corresponding to the feature point $P_i$ is offset by the search center $(\Delta x, \Delta y)$ on the search area of the SAR image.

We expand the square term on the right side of the formula (15) above to obtain:

$$SSD_i(\Delta x, \Delta y) = \sum_{(x,y)} \left( D^i_{Opt}(x,y) \right)^2 + \sum_{(x,y)} \left( D^i_{SAR}(x - \Delta x, y - \Delta y) \right)^2$$
$$-2 \sum_{(x,y)} D^i_{Opt}(x,y) D^i_{SAR}(x - \Delta x, y - \Delta y) \tag{16}$$

The offset $(\Delta x, \Delta y)$ obtained when $SSD_i(\Delta x, \Delta y)$ in the above formula reaches the minimum value is the offset of the matching point relative to the center of the search region. The first term on the right side of the above formula is obviously a constant. In addition, the calculation results of the second term at different positions of the image have little effect on the calculation of the value of the third term [49].

So the term to be solved is:

$$(\Delta x, \Delta y) = \arg \max_{(\Delta x, \Delta y)} \left( \sum_{(x,y)} D^i_{Opt}(x,y) D^i_{SAR}(x - \Delta x, y - \Delta y) \right) \tag{17}$$

In the above formula, the right side of the formula is the correlation operation, and so it can be accelerated by FFT conversion to frequency domain dot product. Therefore, the above formula can be further written as:

$$(\Delta x, \Delta y) =$$
$$\arg \max_{(\Delta x, \Delta y)} \left( \sum_{(x,y)} FT^{-1} \left\{ FT\left[ D^i_{Opt}(x,y) \right] \cdot FT\left[ D^i_{SAR}(x - \Delta x, y - \Delta y) \right]^* \right\} \right) \tag{18}$$

In the above formula, $FT$ and $FT^{-1}$ represent the Fourier transform and inverse transform, respectively. $[\cdot]^*$ represents a complex conjugate. Although the above method can quickly search to obtain the corresponding matching point coordinates on the SAR image, due to the large difference in imaging between optical and SAR images, there are often multiple peaks in the search surface. Mismatches often occur when the largest peak is selected.

In order to more intuitively show the multi-peak phenomenon of the search surface, we select the optical image in Figure 2 and its corresponding SAR image in the same region. The template window position on the optical image and the search region window on the SAR image are shown in Figure 7a,b.

As shown in Figure 7c, at this time, for the template search of the feature point, since there are multiple peaks on the search surface, if the search radius is not set properly, it is easy to generate false matches. Therefore, we consider adding non-maximum suppression to the search surface in the template search step [35].

As shown in Figure 7c, by calculating the ratio of the main and secondary peaks in the map and setting the threshold $T$, when the ratio is greater than $T$, the feature point will be output as a matching point. Due to the interference of the pixels around the main peak, it is impossible to directly select the main and secondary peaks in order on the search

surface map. Therefore, it is necessary to suppress the pixels around the main peak first. The suppression method is shown in Figure 8.

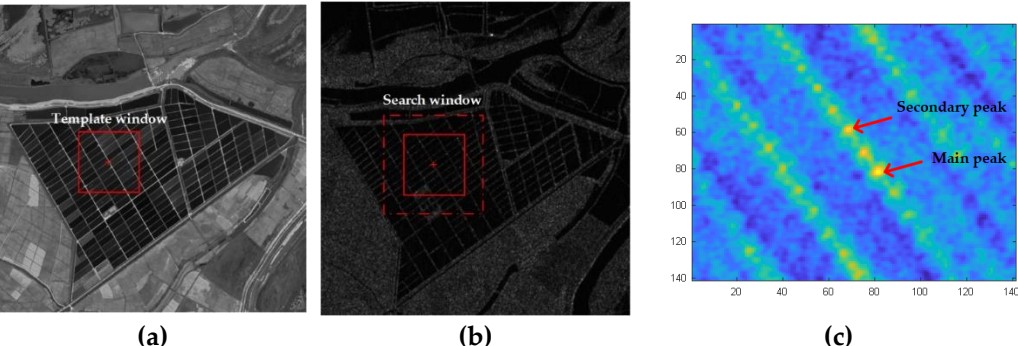

| (a) | (b) | (c) |

**Figure 7.** Similarity surface multi-peak phenomenon. (**a**) Optical image template window; (**b**) SAR image search window; (**c**) search surface map.

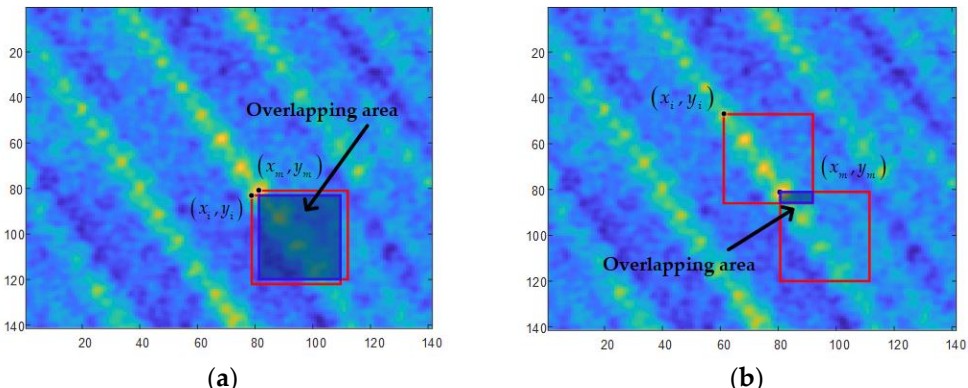

| (a) | (b) |

**Figure 8.** Schematic diagram of non-maximum suppression. (**a**) Non-extreme point that needs to be suppressed; (**b**) extreme point that needs to be preserved.

First, the pixel intensities in the search surface map are sorted from large to small, and the points corresponding to the first $N_s$ values with larger intensities are selected as candidate points. The point with the largest intensity value is taken as the main peak point $(x_m, y_m)$. Among the remaining $N_s - 1$ candidate points, we take each point's coordinates $(x_i, y_i)$ as the upper left corner to construct a square search window with a window width of $w_s$, as shown in Figure 8. Calculate the region overlap ratio of the search window corresponding to the candidate point and the fixed window corresponding to the main peak point, and set the threshold $R_t$. Eliminate the candidate points whose region overlap ratio is greater than $R_t$ (as shown in Figure 8a); otherwise, keep the candidate points (as shown in Figure 8b). After this step, the interference points around the main peak point will be eliminated. Then, the secondary peak point (if any) is found from the remaining candidate points, and the main-to-secondary-peak ratio is calculated. If there is no secondary peak point (it is considered that the ratio of the main and secondary peaks is equal to the set threshold), this point is directly output as a matching point.

Figure 9 shows the non-maximum suppression results of the search surface corresponding to several feature points. The parameters for non-maximum suppression are set as follows: the number of candidate points $N_s$ takes 1% of the number of template pixels; region overlap ratio threshold $R_t = 0.9$; search window size $w_s$ takes the size of the template window; the main-to-secondary-peak ratio $T = 1/0.9$. It can be clearly seen that after non-maximum suppression, the interfering pixels around the main peak are eliminated, and the secondary peak is easier to obtain at this time.

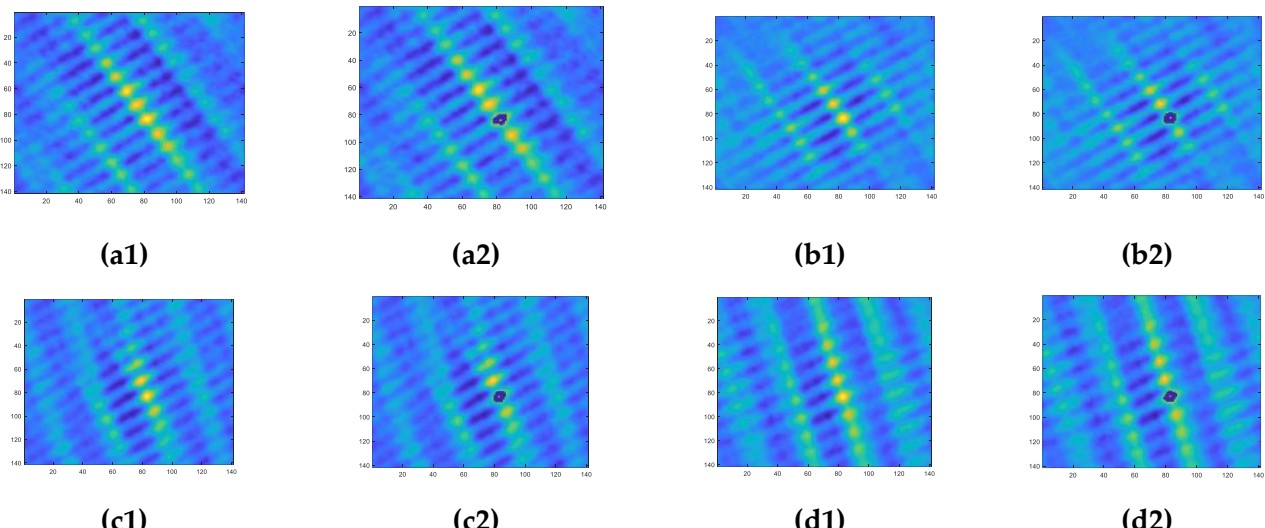

**Figure 9.** Non-maximum suppression results. (**a1**) Point 1 before suppression; (**a2**) Point 1 after suppression; (**b1**) Point 2 before suppression; (**b2**) Point 2 after suppression; (**c1**) Point 3 before suppression; (**c2**) Point 3 after suppression; (**d1**) Point 4 before suppression; (**d2**) Point 4 after suppression.

After completing the template search and matching, the initial corresponding point pairs will be output. At this time, there may still be some mismatched point pairs in the output. Therefore, gross error elimination is still required. The gross error elimination method selected in this paper is the iterative RMSE method, and its processing flow is shown in Figure 10.

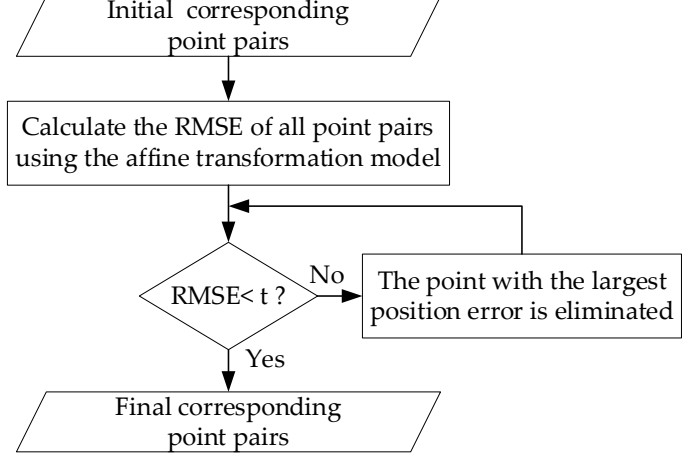

**Figure 10.** Gross error elimination flowchart.

After the gross error elimination step, the final corresponding point pairs are obtained. We use these point pairs to estimate the affine transformation model and then achieve co-registration of optical and SAR images.

## 3. Results

In this section, we use a pair of optical and SAR images to verify the effectiveness of our consistent gradient calculation method. We compare the smoothness of the SSD similarity search surfaces corresponding to SRAWG, CFOG, and HOPES to verify the robustness of our proposed feature description. Then, we analyze the efficiency gains that can be brought about by adopting the template merging strategy. Finally, we select seven pairs of optical and SAR images for a comprehensive registration experiment. The

registration performance of SRAWG, CFOG, HOPES, HOPC, and RIFT is compared by subjective and objective evaluation criteria.

### 3.1. Effectiveness of Consistent Gradient Computation for Optical and SAR Images

In order to verify the advantages of our gradient calculation method compared to the differential gradient operator, we choose the optical and SAR images in Figure 7 for gradient calculation. The processing results of the differential gradient operator, the single-scale Sobel operator, and the ROEWA operator are shown in Figure 11.

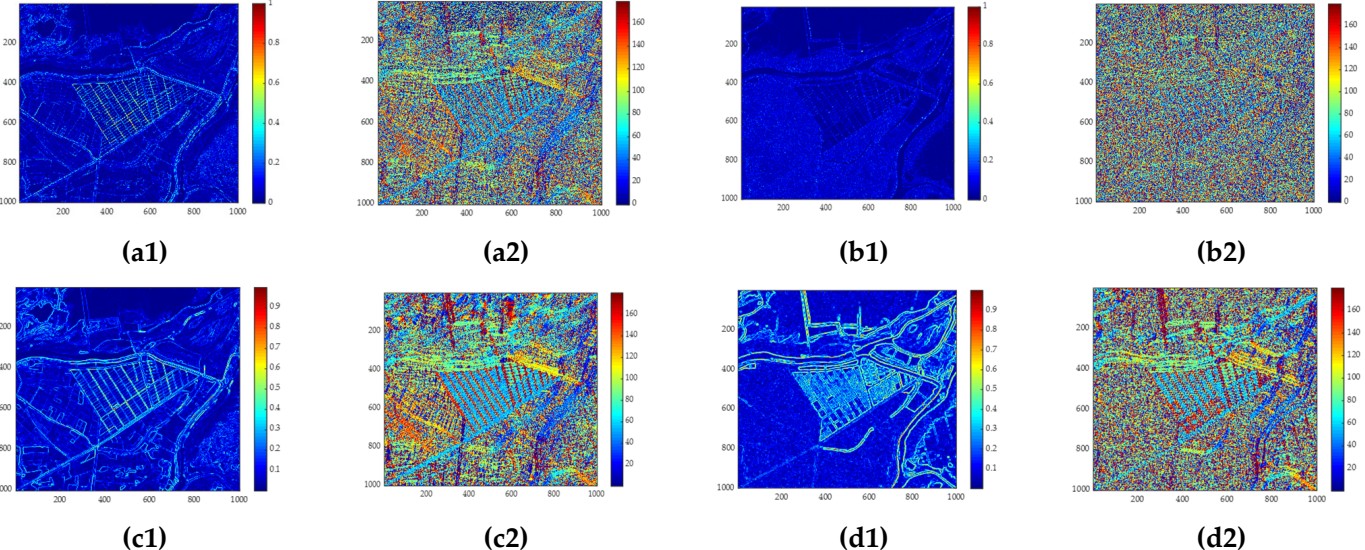

**Figure 11.** Gradient calculation results of differential operator, single-scale Sobel, and ROEWA operators. (**a1**,**a2**) Differential gradient magnitude and direction for optical image; (**b1**,**b2**) differential gradient magnitude and direction for SAR image; (**c1**,**c2**) single-scale Sobel gradient magnitude and orientation for optical image; (**d1**,**d2**) ROEWA gradient magnitude and direction for SAR image.

As can be seen from Figure 11, the differential gradient operator can achieve a good edge detection effect when applied to an optical image. However, when it is applied to a SAR image, due to the interference of speckle noise, the consistency between the edges extracted from optical and SAR images is poor. The extraction results of optical and SAR images using single-scale Sobel and ROEWA operators have good consistency in gradient magnitude and direction. This further shows that the use of single-scale Sobel and ROEWA operators instead of a difference operator can obtain more consistent gradients.

### 3.2. Comparison of SSD Search Surfaces with Different Feature Descriptions

To verify the advantages of the SRAWG description, we extract a feature point on each of the three pairs of optical and SAR images. The 3D dense feature description of the region around the feature point is constructed by three descriptions: SRAWG, CFOG [6], and HOPES [7], respectively. The SSD search surfaces are calculated by template matching. The experimental results are shown in Figure 12.

From Figure 12, we can see that the search surface obtained by the SRAWG description is the smoothest compared to the CFOG and HOPES descriptions. The search surfaces obtained by the latter two have multiple peaks and are not smooth. The optical and SAR image descriptions constructed by SRAWG have better similarity.

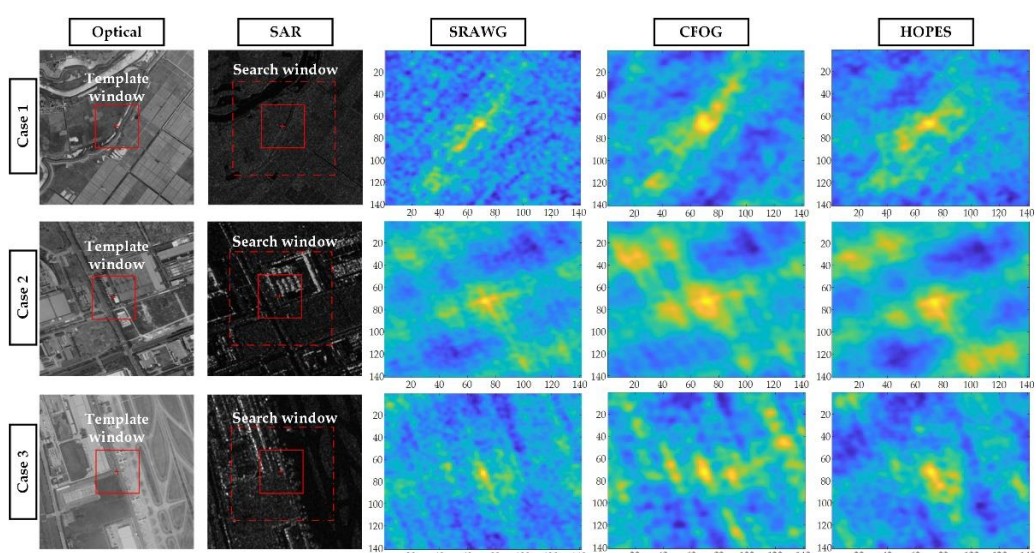

**Figure 12.** SSD search surfaces for three descriptions.

### 3.3. Overlapping Template Merging Performance Analysis

In order to intuitively display the effect of overlapping template merging processing, we select an optical image for processing and compare the overlap of each template before and after template merging. The obtained result is shown in Figure 13 below.

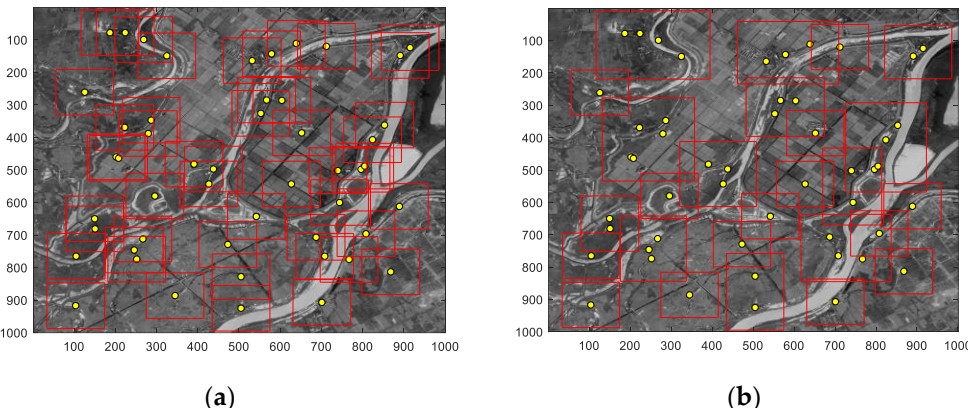

(**a**)                                                                                 (**b**)

**Figure 13.** Template merging effect. (**a**) Before merging; (**b**) after merging. The yellow points are feature points, and their corresponding red boxes represent their respective template areas.

Comparing Figure 13a,b, it can be found that after the template merging operation, the overlapping degree of the templates corresponding to the feature points is greatly reduced compared with that before the merging. In order to quantitatively study the ability of template merging to improve the efficiency of feature construction, we designed the following experiment: As mentioned earlier, when the number of feature points to be extracted remains unchanged, as the image size becomes smaller, the template overlap will become more and more serious. This change rule can also be equivalent by keeping the image size unchanged and adjusting the number of feature points that need to be extracted from the image. Therefore, we choose an optical image in Figure 13 and its corresponding SAR image for the experiment so that the number of feature points extracted on the image gradually increases, and then the reduction rate of image registration time after template merging processing is calculated. The experimental result is shown in Figure 14.

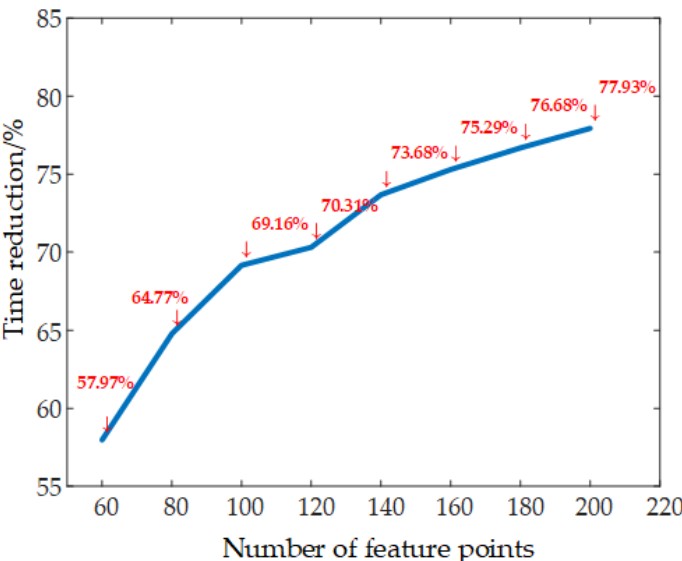

**Figure 14.** Time reduction rate curve after template merging.

As can be seen from Figure 14, when the image size remains unchanged, as the number of feature points extracted on the image continues to increase, the time reduction rate after template merging processing also continues to rise. This further validates the conclusion we gave earlier.

### 3.4. Image Registration Experiments

#### 3.4.1. Experimental Data and Parameter Settings

To verify the effectiveness of our proposed registration method, we select optical and SAR images from different sensors and different resolutions. There is a certain time difference between each pair of images, and the topographic features of the areas covered by different images are also different. The details of each pair of images are shown in Table 1, and the corresponding images are shown in Figure 15. In addition, when the central incidence angle of SAR is different, the degree of geometric difference between the corresponding SAR image and the optical image is also different. Therefore, we give the central incidence angles corresponding to each pair of images in Table 1. Optical and SAR images can undergo a coarse co-registration process with their respective sensor parameters (RPC or RD model), which transforms the two images from their respective raw data to a unified geographic coordinate system. Since optical and SAR images may differ in their original spatial resolutions, we need to resample both images to the same ground sample distance (GSD) in the coarse registration stage. There are still small rotation and scale differences between images after coarse registration. There is also a large amount of translation between images. The experiment compared SRAWG, CFOG, HOPC, HOPES, and RIFT. The parameter settings of each method are shown in Table 2.

**Table 1.** Optical and SAR images.

| Case | Dataset Description | | |
|---|---|---|---|
| | **Reference Image** | **Sensed Image** | **Image Characteristic** |
| 1 | Sensor: Google Earth<br>Original Resolution: 5 m<br>Central Incidence Angle: 0°<br>Size: 1000 × 1000<br>GSD: 5 m | Sensor: GF-3<br>Original Resolution: 10 m<br>Central Incidence Angle: 24.71°<br>Size: 1000 × 1000<br>GSD: 5 m | The image covers the river region on the outskirts, and the terrain is relatively flat. The edge of the river is clear on the SAR image, and the speckle noise is relatively weak. However, there are significant nonlinear intensity differences compared to optical image. |

**Table 1.** *Cont.*

| Case | Dataset Description | | |
|---|---|---|---|
| | Reference Image | Sensed Image | Image Characteristic |
| 2 | Sensor: Google Earth<br>Original Resolution: 1 m<br>Central Incidence Angle: 0°<br>Size: 1000 × 1000<br>GSD: 3 m | Sensor: GF-3<br>Original Resolution: 3 m<br>Central Incidence Angle: 43.28°<br>Size: 1000 × 1000<br>GSD: 3 m | The image covers an urban region with tall buildings and has local geometric distortions. There is significant speckle noise on SAR images. |
| 3 | Sensor: Google Earth<br>Original Resolution: 5 m<br>Central Incidence Angle: 0°<br>Size: 1500 × 1500<br>GSD: 10 m | Sensor: GF-3<br>Original Resolution: 10 m<br>Central Incidence Angle: 20.23°<br>Size: 1500 × 1500<br>GSD: 10 m | The image covers the airport region, which has clear geometry in both images. There is a clear intensity inversion on the SAR image and optical image, and significant speckle noise on the SAR image. |
| 4 | Sensor: Google Earth<br>Original Resolution: 1 m<br>Central Incidence Angle: 0°<br>Size: 1500 × 1500<br>GSD: 1 m | Sensor: Airborne SAR<br>Original Resolution: 0.5 m<br>Central Incidence Angle: 81.24°<br>Size: 1500 × 1500<br>GSD: 1 m | The image covers suburban regions with complex structures such as buildings, reservoirs, farmland, etc. On the SAR image, some shadow areas are generated due to the large incident angle, and the image has some defocusing phenomenon. |
| 5 | Sensor: Google Earth<br>Original Resolution: 1 m<br>Central Incidence Angle: 0°<br>Size: 1500 × 1500<br>GSD: 1 m | Sensor: Airborne SAR<br>Original Resolution: 1 m<br>Central Incidence Angle: 80.89°<br>Size: 1500 × 1500<br>GSD: 1 m | The image covers a large region of rural farmland with relatively flat terrain. Compared with the optical image, there is a significant intensity inversion phenomenon in the SAR image. |
| 6 | Sensor: Google Earth<br>Original Resolution: 5 m<br>Central Incidence Angle: 0°<br>Size: 2570 × 1600<br>GSD: 10 m | Sensor: Sentinel-1<br>Original Resolution: 10 m<br>Central Incidence Angle: 38.94°<br>Size: 2570 × 1600<br>GSD: 10 m | The image covers the outskirts of a city, which is flat and has complex features such as buildings, farmland, lakes, and road networks. |
| 7 | Sensor: Google Earth<br>Original Resolution: 5 m<br>Central Incidence Angle: 0°<br>Size: 1274 × 1073<br>GSD: 10 m | Sensor: Sentinel-1<br>Original Resolution: 10 m<br>Central Incidence Angle: 39.13°<br>Size: 1274 × 1073<br>GSD: 10 m | The image covers the hilly area around the river. The terrain in this area is greatly undulating, and the geometric difference between the optical image and the SAR image is large. |

**Table 2.** The experimental parameter settings of each method.

| Methods | Parameter Space |
|---|---|
| SRAWG | $N = 5, k = 8, N = 9, \sigma = 0.8, \alpha_n = 2, \beta_m = 2, R_t = 0.9, T = 1/0.9,$<br>$N_s = 1\%$ number of template pixels, Template size: 100 × 100,<br>Radius of search region: 20 pixels |
| CFOG | $n = 5, k = 8, N = 9, \sigma = 0.8$, Template size: 100 × 100, Radius of<br>search region: 20 pixels |
| HOPC | $n = 5, k = 8, N = 9$, Template size: 100 × 100, Radius of search<br>region: 20 pixels |
| HOPES | $n = 5, k = 8, N = 9, \sigma = 0.8, N_{scale} = 3, \sigma_{MSG} = 1.4, k_{MSG} = 1.4,$<br>$\gamma_{MSG} = 6, R_t = 0.9, T = 1/0.9, N_s = 1\%$ number of template pixels,<br>Template size: 100 × 100, Radius of search region: 20 pixels |

To be fair, for SRAWG, CFOG, HOPES, and HOPC with the same registration framework, we use block–Harris to complete feature point extraction, and the total number of extracted feature points is $n \times n \times k$. Since RIFT is a SIFT-like method, we perform feature point extraction as suggested by the authors in the original article. All methods use the iterative method proposed in this paper to complete the gross error elimination process. The threshold value *t* of the gross error elimination is set to 1.5 pixels. The rest of the parameter settings of the compared methods are selected as the parameters recommended by the authors in their original articles. For details, please refer to their articles. All methods are implemented based on MATLAB, and the code of HOPC is provided by the authors of the original article.

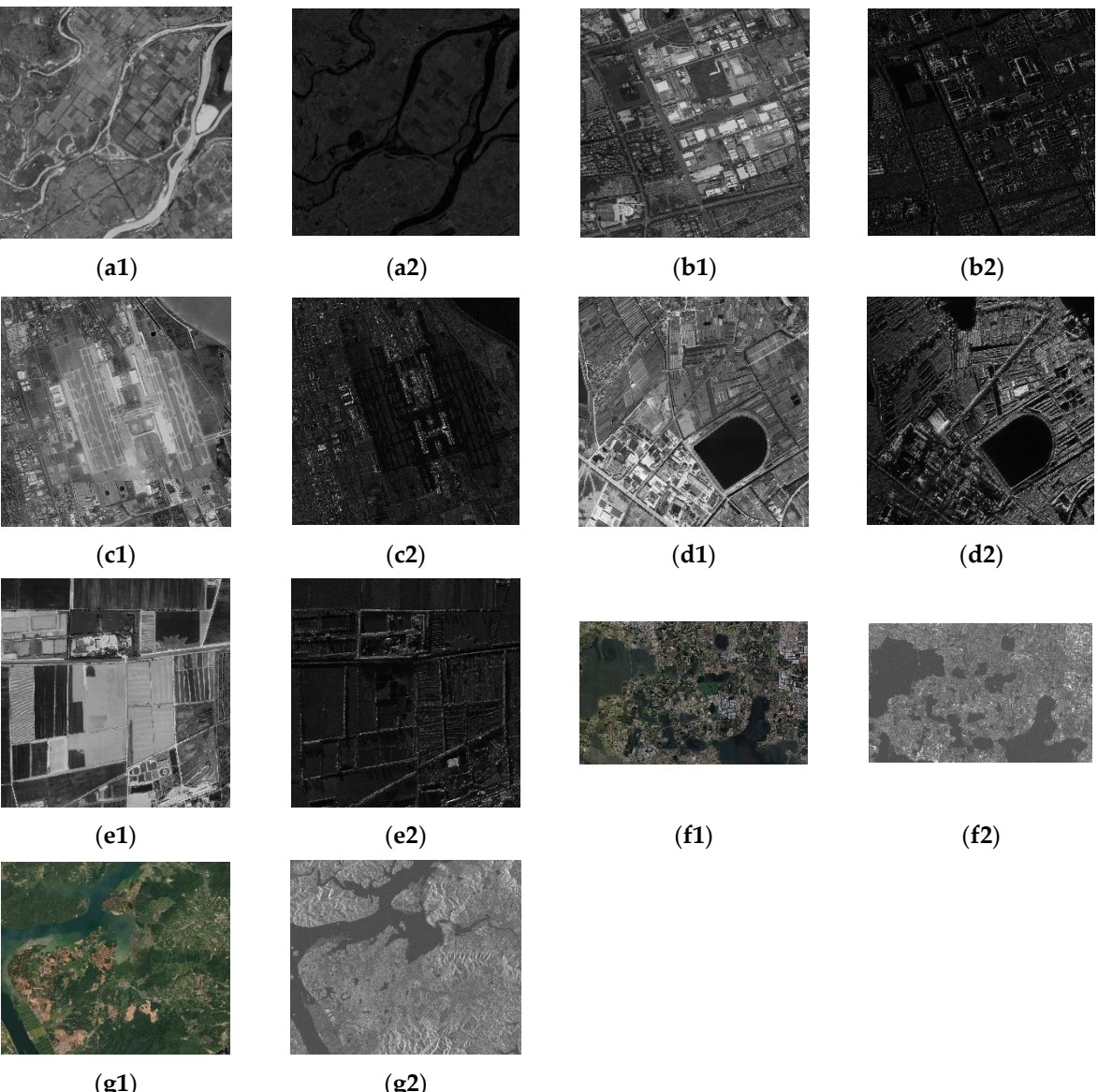

**Figure 15.** Optical and SAR images. (**a1–g1**) Optical images; (**a2–g2**) SAR images.

3.4.2. Evaluation Criteria

(1) Subjective evaluation: mosaic map, by superimposing the two registered images into grids, and by magnifying the boundary between the grids, observe the geometric stitching quality of the two images.

(2) Objective evaluation:

1. Number of Correct Matches (NCM)

$$\text{NCM} = \sum_{i=1}^{M} \left( \| H(x_r{}^i, y_r{}^i) - (x_s{}^i, y_s{}^i) \|_2 < 1.5 \right) \tag{19}$$

In the above formula, $(x_r{}^i, y_r{}^i)$ and $(x_s{}^i, y_s{}^i)$ represent the corresponding point pair on the reference image and the sensed image, respectively, which are output after the gross error elimination algorithm. $H$ represents the transformation model from the reference image to the sensed image, which is calculated by manually selecting 50 pairs of checkpoints evenly distributed on the reference image and the sensed image. Among all output corresponding point pairs $M$, the number of point pairs whose distance is less than the threshold of 1.5 pixels after $H$ transformation is calculated as NCM.

2. Correct Match Rate (CMR)

$$\text{CMR} = \frac{\text{NCM}}{M} \tag{20}$$

In the above formula, $M$ is the number of corresponding point pairs output after gross error elimination.

3. Root Mean Squared Error (RMSE)

$$\text{RMSE} = \frac{1}{M} \sum_{i=1}^{M} \left\| H\left(x_r^i, y_r^i\right) - \left(x_s^i, y_s^i\right) \right\|_2 \tag{21}$$

4. Run Time

Count and compare the time consumed by each method from feature extraction to feature matching.

### 3.4.3. Results and Analysis

The registration results of the seven pairs of images corresponding to the five methods are shown in Figure 16.

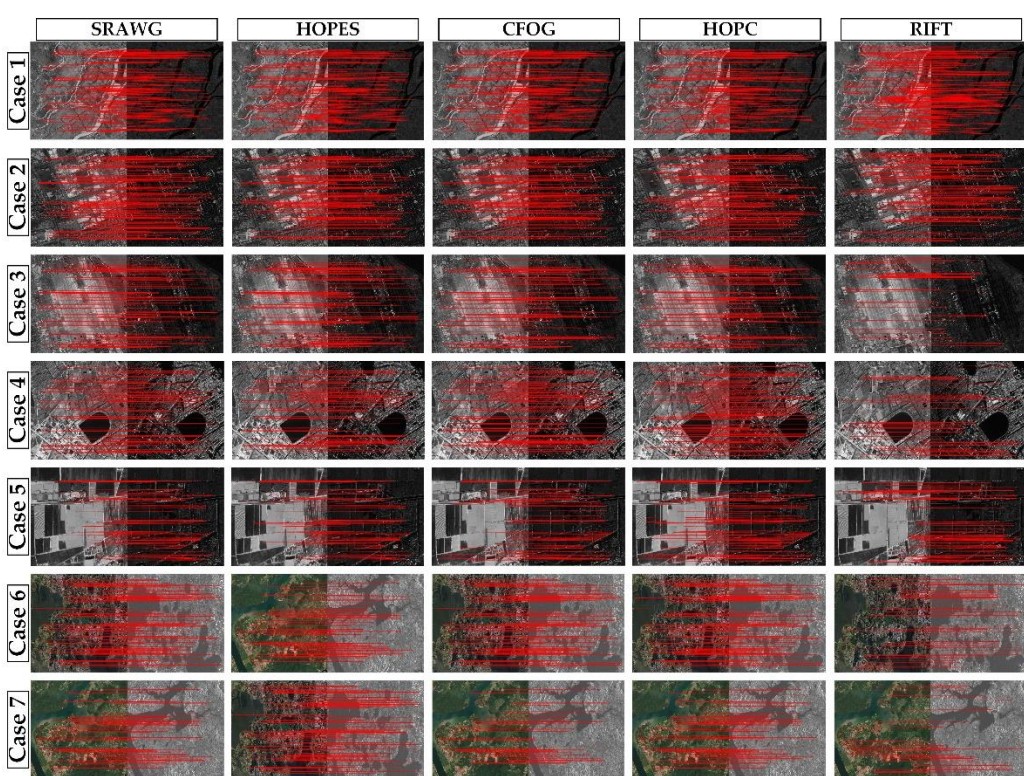

**Figure 16.** Registration results.

It can be seen from the registration results in Figure 16 that each method can obtain considerable and uniformly distributed corresponding point pairs. Because the RMSE gap between them is not very large, it is difficult to observe the difference in their splicing effects from the mosaic map. In this paper, we only show the mosaic map result of the second pair of images with significant differences in splicing effect (as shown in Figure 17).

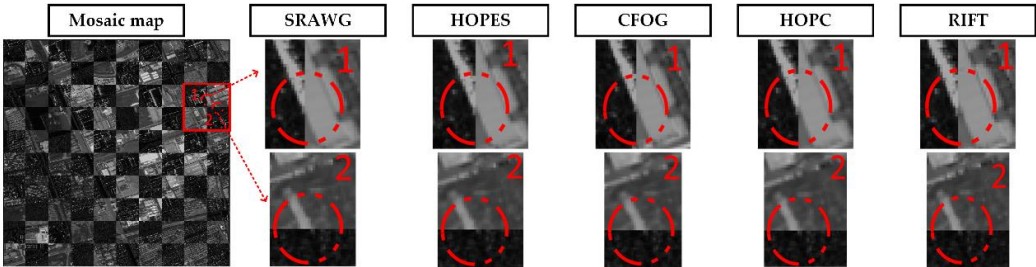

**Figure 17.** Mosaic map for Case 2. The geometric stitching effect of the two images can be observed from the center of the area circled by circles 1 and 2.

In Figure 17, SRAWG produces the smallest shift in the two red circled regions. The splicing offsets of HOPES, HOPC, and RIFT in the red circle area 1 are larger than those of SRAWG and CFOG. The splicing offsets of CFOG, HOPC, and RIFT in the red circle area of No. 2 are larger than those SRAWG and HOPES. To quantitatively compare the registration performance of the five methods, we count their respective NCM, CMR, RMSE, and run time (in seconds), as shown in Table 3.

**Table 3.** NCM, CMR, RMSE, and run time of four methods. For each pair of experimental images, four metrics are used to measure the algorithm being compared. The numbers labeled in bold represent the optimal values for all algorithms under this metric.

| Case | Criteria | SRAWG | HOPES | CFOG | HOPC | RIFT |
|------|----------|-------|-------|------|------|------|
| 1 | NCM | 177 | **178** | 156 | 167 | 147 |
| | CMR | 90.31 | **91.28** | 79.59 | 84.34 | 51.22 |
| | RMSE | **0.9257** | 1.0170 | 1.2618 | 1.4919 | 1.7086 |
| | Run Time | **5.5** | 16.4 | 6.1 | 43.68 | 12.99 |
| 2 | NCM | **171** | 139 | 159 | 145 | 92 |
| | CMR | **86.36** | 71.65 | 80.3 | 73.23 | 49.73 |
| | RMSE | **1.0413** | 1.4455 | 1.4143 | 1.4632 | 1.8324 |
| | Run Time | **5.6** | 16.5 | 6.2 | 43.01 | 12.74 |
| 3 | NCM | **166** | 152 | 118 | 106 | 14 |
| | CMR | **90.71** | 85.88 | 67.05 | 66.67 | 28 |
| | RMSE | **0.9109** | 1.0575 | 1.5540 | 1.5148 | 2.5998 |
| | Run Time | 8.3 | 16.8 | **6.2** | 48.59 | 16.39 |
| 4 | NCM | 112 | 96 | **114** | 103 | 13 |
| | CMR | **68.71** | 68.57 | 65.9 | 60.95 | 15.85 |
| | RMSE | 1.9126 | **1.7773** | 2.0346 | 2.1014 | 4.5432 |
| | Run Time | 8.2 | 16.6 | **6.3** | 48.86 | 17.21 |
| 5 | NCM | 92 | 89 | 62 | 99 | 39 |
| | CMR | **92.93** | 85.58 | 59.62 | 68.28 | 41.05 |
| | RMSE | **0.9936** | 1.1819 | 1.6501 | 1.5363 | 1.8114 |
| | Run Time | 7.8 | 16.6 | **6.2** | 48.64 | 16.93 |
| 6 | NCM | **183** | 175 | 163 | 171 | 34 |
| | CMR | **95.31** | 91.15 | 89.07 | 88.14 | 38.2 |
| | RMSE | **0.7376** | 0.9189 | 1.0536 | 1.3901 | 2.0999 |
| | Run Time | 9.70 | 16.92 | **6.34** | 53.41 | 22.23 |
| 7 | NCM | **72** | 66 | 51 | 42 | 15 |
| | CMR | **60** | 45.52 | 58.62 | 34.71 | 25 |
| | RMSE | **1.8506** | 2.2699 | 2.0861 | 2.2457 | 2.5719 |
| | Run Time | 6.38 | 15.48 | **5.66** | 48.09 | 14.75 |

As can be seen from Table 3, our proposed SRAWG method achieves the smallest RMSE and the highest CMR on six pairs of images, HOPES is second only to SRAWG, and the worst is RIFT. In terms of registration efficiency, it can be seen that when the number of

feature points to be extracted on the image is fixed, the processing efficiency of SRAWG is better than that of CFOG when the image size is relatively small. This also benefits from the advantage of adopting the template merging strategy, but this advantage cannot be maintained with the increase in image size. This is because as the size of the image increases, the degree of overlap between templates also decreases, and the improvement brought by the template merging strategy becomes smaller and smaller. Therefore, CFOG has the highest processing efficiency, followed by SRAWG, HOPES, and HOPC. On NCM, there is no method that can achieve an overwhelming advantage. However, it is not difficult to find that on each pair of images, the gap between SRAWG's NCM and the largest NCM is the smallest. Compared with CFOG, which is the most efficient in registration, our method has an average improvement of about 26% in accuracy in the case of slightly weaker efficiency. Compared with the HOPES method with the highest registration accuracy among the compared methods, our method improves the registration efficiency by an average of about double and improves the accuracy by about 14%.

HOPC improves robustness to NID between optical and SAR images by constructing phase-consistent histograms with radiation and contrast invariance. The experimental results show that it can indeed successfully match seven pairs of optical and SAR images, but its accuracy is not the best among the five methods. In addition, due to the complexity of phase consistency calculation, its registration efficiency is the lowest among the five methods. It also shows that this structural feature extraction method is not optimal for the co-registration of optical and SAR images.

In CFOG, the feature description is constructed by obtaining the horizontal and vertical gradients of the image through the differential gradient operator, and the gradient in other directions is obtained by sinusoidal interpolation. Although this description construction method is simple and efficient (in fact, its efficiency is the highest among the five methods), due to the existence of the superior speckle noise in the SAR image, it is easy to produce a big difference from the optical image in the initial horizontal and vertical gradient calculation. Therefore, its registration accuracy is significantly weaker than that of the SRAWG and HOPES methods.

HOPES directly constructs feature description through multi-scale and multi-directional Gabor filters. Then, multi-scale feature fusion is performed through the main edge fusion strategy to enhance the main edge features. This also makes it have higher registration accuracy compared to CFOG, HOPC, and RIFT. Due to the utilization of multi-scale information, its registration efficiency is also significantly lower than that of SRAWG and CFOG.

RIFT performs mixed detection of corner points and edge points on the phase consistency map to complete feature point extraction. It builds the maximum index map (MIM) through the Log-Gabor convolution sequence, then uses the MIM to build the feature description, and finally completes the pairing of the feature descriptor by calculating the Euclidean distance between the descriptors. Although this method can successfully register all image pairs, its registration accuracy is the worst among the compared methods. Because it needs to calculate the phase consistency of the image and needs to compare the Euclidean distance between vectors one by one in the feature matching stage, the efficiency of its registration is not high.

SRAWG uses single-scale Sobel and ROEWA operators to calculate the consistent gradients of optical and SAR images, respectively, which obtains good gradient consistency. In order to enhance the robustness of the description, the angle-weighted gradient method is used to construct the feature description pixel by pixel, and the template merging strategy is used to further improve the matching efficiency in template matching. Therefore, it can achieve efficient and high-precision registration.

## 4. Discussion

In Section 3, we verify that the proposed method achieves the highest registration accuracy while maintaining efficiency by performing co-registration experiments on multiple pairs of real optical and SAR images. Now we analyze the effect of relevant parameter

settings on the registration performance. The key parameters to be studied are the template size, the neighborhood of feature construction, and the number $N$ of feature directions. As the values of these three parameters continue to increase, the registration efficiency of the method gradually decreases. We should choose appropriate parameter settings to achieve the compromise between accuracy and efficiency. Under different parameter settings, we calculate the mean value of RMSEs for all image pairs in Table 1 as the mean error. The effect of different parameter settings on the accuracy is shown in Figure 18.

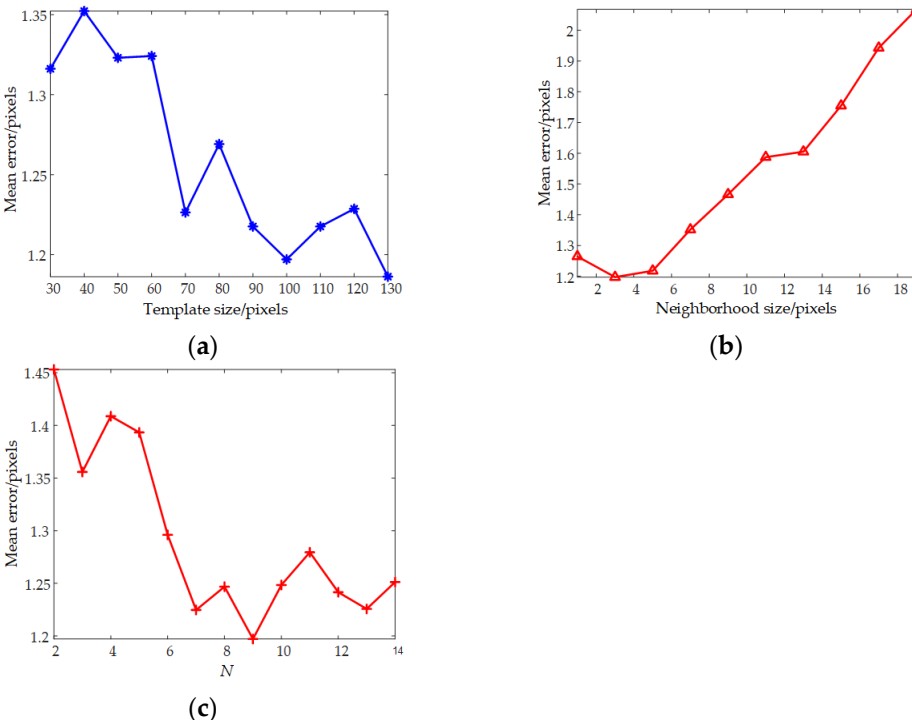

**Figure 18.** The effect of parameter settings on the mean error. (**a**) Template size changes; (**b**) feature neighborhood size changes; (**c**) feature directions.

We first analyze the influence of template size on registration accuracy. We fix $N = 9$, and the neighborhood of feature construction is $3 \times 3$. When the template size is changed from $30 \times 30$ to $130 \times 130$ at intervals of 10 pixels, the obtained mean error curve is shown in Figure 18a. It is not difficult to find that as the template size increases, the mean error generally shows a downward trend. The improvement rate of the registration accuracy decreases significantly after the template size is $70 \times 70$. Therefore, considering the registration efficiency, the template size should not be too large. Then, we analyze the effect of the neighborhood size constructed by the feature description on registration accuracy. We fix the template size as 100 and $N = 9$ and plot the corresponding mean error curve when the neighborhood size varies from $1 \times 1$ to $19 \times 19$, as shown in Figure 18b. It can be seen that the minimum mean error is achieved when the neighborhood size is $3 \times 3$, and then increasing the neighborhood size will cause the error to increase. This may be due to more interference introduced in the feature description as the neighborhood size increases. Finally, we analyze the influence of the number of feature directions $N$ on registration accuracy. We set the template size to be $100 \times 100$ and the neighborhood of feature construction to be $3 \times 3$ and plot the mean error curve between $N$ from 2 to 14, as shown in Figure 18c. Obviously, when the value of $N$ increases to a certain extent ($N = 9$), the improvement of the registration accuracy is very small.

The number of feature points to be extracted In the feature extraction stage needs to be adjusted appropriately according to the actual image size. When the image size is large, the number of feature points to be extracted can be appropriately increased, but it should be noted that an excessive number of feature points may bring a greater computational burden.

When the image we want to register contains many weak texture areas (such as water surface), the number of blocks should not be too large; otherwise, some unreliable feature points may be detected in these areas. In the feature matching stage, the larger the number of candidate feature points $N_s$ is set, and the smaller the search window area overlap ratio threshold $R_t$, the larger the suppressed range around the main peak. Increasing the main-to-secondary-peak ratio threshold reduces the number of corresponding point pairs in the final output.

Now we summarize the limitations of the proposed method. The method proposed in this paper needs to use sensor parameters to complete the coarse registration before registration. In practical applications, there will still be a small part of data that cannot use sensor parameters to complete the coarse registration process, and so the scope of application of this method will be limited to a certain extent.

## 5. Conclusions

In this paper, we propose a fast registration method based on SRAWG feature description. The main innovations of the method are as follows: (1) In order to improve registration accuracy, we first propose the SRAWG 3D dense feature description by combining the consistent gradient extraction and angle-weighted gradient. Second, for the multi-peak problem of the search surface in template search, we introduce a non-maximum suppression method to improve the accuracy of the feature matching. (2) In order to maintain a high registration efficiency, we first modify the original multi-scale Sobel and ROEWA operators to the single-scale case. Second, we introduce an overlapping template merging strategy to further improve the efficiency of feature description. Co-registration experiments are carried out on multiple pairs of optical and SAR images covering different regional features. In experiments, we compare the state-of-the-art CFOG, HOPES, HOPC, and RIFT methods. Compared with the most efficient CFOG, although the efficiency is slightly lower, the accuracy is improved by about 26% on average. Compared with HOPES with the highest accuracy, the proposed method not only improves the efficiency by an average of about double but also improves the accuracy by an average of about 14%.

**Author Contributions:** All the authors made significant contributions to the work. Z.W. and A.Y. carried out the experiment framework. Z.W., A.Y., X.C. and B.Z. processed the experiment data; Z.W. wrote the manuscript; Z.W. and A.Y. analyzed the data; Z.D. gave insightful suggestions for the work and the manuscript. All authors have read and agreed to the published version of the manuscript.

**Funding:** This research was funded partly by the National Natural Science Foundation of China (NSFC) under Grant NO. 62101568 and partly by the Scientific Research Program of the National University of Defense Technology (NUDT) under Grant ZK21-06.

**Data Availability Statement:** The data presented in this study are available on request from the corresponding author.

**Acknowledgments:** The authors would like to thank all the anonymous reviewers for their valuable comments and helpful suggestions which led to substantial improvements in this paper.

**Conflicts of Interest:** The authors declare no conflict of interest.

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
