# Peer review of "A Fast Registration Method for Optical and SAR Images Based on SRAWG Feature Description"

_remotesensing, doi:10.3390/rs14195060_

Round 1

Reviewer 1 Report

This paper proposes a novel image co-registration method for matching SAR and optical imageries. The basic strategy of the proposal is to extract feature points intensively and thus, sounds somehow a combination and modification of the existing methods. This itself is not a problem however; on the other hand, the experimental setups and the results seem the proposal not superior than the existing ones.

As long as this reviewer understands from Section 2, the proposal maximizes the RMSE score in the last part of the algorithm. That is, the proposal optimizes the parameters while this process is not always applied by the others and thus, it might contribute a lot in the experiments. In addition, this process may work if the users know the correct registration results but this is not always the case. That is, the proposal lacks generality or not applicable for the real use case.

Experimental setups are too limited and lack generality. There seems no discussion for the fore-shortening. As the relationship between the incidence angle and skew is opposite for optical and radar observations, the experiments must include such features. In addition, there seems no discussion if two imageries have different spatial resolutions.

In table 3, the proposal is not always superior to the existing ones. The difference of the scores is close and thus, it seems that the proposal is equivalent to the existing ones but not superior than. If there is. the authors clearly explain and discuss.

Table 4 is useless because this is nothing but an opinion of the authors. More meaningful information should be added, for example, superiority in noise, resolution difference, resilience against the image skew etc. Something original and new should be exist in the proposal.

Author Response

We have carefully revised the manuscript according to your suggestions, please refer to the attachment for details.

Reviewer 2 Report

This manuscript presents an SAR and optical image registration method using the orientation histogram of SOBEL and ROEWA features (SRAWG). Generally, the manuscript is well written, and the experiments show the outperformance of the proposed method. However, there are still some issue should be considered to improve the manuscript.

1. In table 2, why the “search region” of HOPC is 10 pixels, smaller than the other descriptors?

2. In SRAWG, both SOBEL and ROEWA are used. I think that because SOBEL is well suited for optical images and ROEWA is for SAR images. However, in CFOG, HOPC and HOPES, did the authors only used one type of feature descriptor for both optical and SAR images?

3. The authors should compare with other optical-SAR image registration methods.

4. How many orientations in SRAWG (N=?), as shown in Fig.5. How about the performance when N changes?

5. What will happen if there is small rotation between the input optical and SAR images?

Author Response

(The authors gave the same response as above.)

Reviewer 3 Report

Remarks:

The article is well written. It distinguishes with good analytical descriptions and convinced illustrations.

Remarks:

Disclose the abbreviation ROEWA where it first time appears: ratio of exponentially weighted averages (ROEWA)

The text in the figures is of small size letters. It is not very convenient to read it.

In point of reviewer’s view, the size of the variables in the text and in expressions (numbered formulas) is different.

Figure 11 and its description (capture) are on different pages.

Author Response

(The authors gave the same response as above.)

Reviewer 4 Report

I found this manuscript to be quite well written with interesting ideas in support of an optical and SAR image registration solution.
The whole is described in sufficient detail to appreciate the underlying ideas and their implementation.
An experimental comparison between five pairs of images from GoogleEarth, GF-3, Airborne SAR sensors allows to illustrate quite well the efficiency of the proposed method on different image contents (urban, suburban, rural, ...). The analysis provided looks quite reasonable.

1) What is currently missing is perhaps a more objective description of the limitations of the proposed method, with a clear and complete listing of the important parameters to be set appropriately by the user to obtain acceptable results.

2) For example, why limit the calculation of the attributes to a 3x3 neighborhood (as illustrated in figure 6)? What can be expected with a larger neighborhood size?

3) How to correctly select the threshold t for gross error removal depending on the image and sensor pair ?

4) Why resort to a manual selection of check-points evenly distributed on the reference image and the sensed image to estimate the H-transform? What happens in case of selection errors leading to a slightly or highly inaccurate estimate? What is then the impact on the efficiency of the proposed method?

5) How to appreciate the geometric stitching quality of the two images should be better described as it remains abstract at the moment?

6) What is a significant difference in splicing effect (as explained in figure 17)?

7) From a more general perspective, a discussion section delineating objectively the strengths, weaknesses and limitations of the proposed approach would be valuable.

8) Similarly, testing on other sensor pairs would also add more strength to the comments and conclusions drawn here.

9) Note that some sentences need to be refined.

10) Red circular zones would benefit from enlargement in figure 17.

Typo(s):
The parameters .... are set to:
Through performing registration experiment on multiple pairs of optical and SAR images covering different regional features.

Author Response

(The authors gave the same response as above.)

Round 2

Reviewer 1 Report

The paper seems fine to be published.

Reviewer 4 Report

As far as I am concerned, the authors have answered correctly the questions asked and suggestions made.
The content of the paper has been revised and enriched in different places.

The overall content seems to me to be acceptable for publication.